# The Relationship between Meaning in Life and Mental Health in Chinese Undergraduates: The Mediating Roles of Self-Esteem and Interpersonal Trust

**DOI:** 10.3390/bs14080720

**Published:** 2024-08-16

**Authors:** Benyu Zhang, Anna Wang, Yuan Ye, Jiandong Liu, Lihua Lin

**Affiliations:** 1School of Psychology, Fujian Normal University, Fuzhou 350117, China; zhangbenyu@fjnu.edu.cn; 2Psychological Rehabilitation Center, Fuzhou Neuro-Psychiatric Hospital, Fuzhou 350008, China; wan13030925913@163.com; 3College of Foreign Languages, Fujian Normal University, Fuzhou 350007, China; yeyuan@fjnu.edu.cn; 4School of Health, Fujian Medical University, Fuzhou 350122, China; ljdpsy@fjmu.edu.cn

**Keywords:** mental health, meaning in life, self-esteem, interpersonal trust

## Abstract

To explore the association and the underlying process between meaning in life and psychological health, a stratified random sampling was conducted on undergraduate students from five universities in Fujian Province from March to April 2022, with the Meaning in Life Questionnaire, the Self-Esteem Scale, the Interpersonal Trust Scale, and the Kessler10 Scale. The results indicated that 34.5% of Chinese undergraduates were in poor or worse mental health. There were significant positive correlations among meaning in life, self-esteem, and interpersonal trust; meaning in life, self-esteem, and interpersonal trust were all significantly and positively correlated with mental health. Self-esteem and interpersonal trust played a chain mediating role between meaning in life and mental health. Schools and families should conduct appropriate activities to help them enhance meaning in life so as to improve the level of mental health.

## 1. Introduction

In recent years, the pressure from multiple factors such as the ever-changing living environment and lifestyle, intense study and work, heavy financial burden, and complex and severe employment situation have led to the frequent occurrence of mental health problems among undergraduates [1]. The mental health problems of undergraduates are closely related to the low meaning in life among the college population [2]. The meaning in life is a philosophical concept that has been introduced into the social sciences by psychologists. Steger defines the sense of meaning as the individual’s understanding and awareness of the meaning of his or her own life and the recognition of his or her purpose and mission in life [3]. He proposes that meaning in life consists of two dimensions: the presence of meaning and the search for meaning. As a positive protective factor for mental health, meaning in life has received wide attention. When individuals truly recognize their own values and goals in life, it gives them a strong incentive to work toward self-actualization [4], which, in turn, affects their psychological state and behavioral habits. Studies have shown that meaning in life can increase an individual’s life and academic satisfaction [2,5], and can also enhance the levels of self-esteem in adolescents [6]. Individuals with high meaning in life are more likely to have a higher sense of self-worth, clearer purposes and goals, more direction, confidence, and motivation, and a greater likelihood of academic and professional achievements [7]. Previous research has shown that meaning in life positively predicts an individual’s mental health [3]. Thus, it is evident that meaning in life has an important impact on the mental health of individuals.

Self-esteem may play an important role in the relationship between meaning in life and an individual’s mental health. Self-esteem refers to an individual’s perception of their own worth, which includes not only acknowledging and evaluating oneself and others, but also the emotional reaction to that evaluation [8]. On the one hand, the level of self-esteem can predict an individual’s mental health. Research has suggested that mental health and self-esteem are significantly correlated with one another, with high self-esteem being associated with better mental health [9]. In a cross-cultural study, Diener et al. [10] found that undergraduates’ self-esteem reached above the moderate correlation with life satisfaction. Additional work has also revealed that self-esteem positively predicts undergraduates’ subjective happiness, containing dimensions such as life satisfaction and positive affect, and the subjective happiness of an individual has been regarded as a key determinant of mental health [11]. On the other hand, self-esteem may be influenced by meaning in life. Meaning in life contributes positively to the development of individual self-esteem [12]. First, meaning in life may lead individuals to recognize their value and experience self-esteem; second, meaning in life may lead individuals to better obtain goals, which, in turn, may enhance self-esteem. Therefore, meaning in life may influence individual’s self-esteem and may influence mental health through self-esteem.

**Hypothesis** **1** **(H1):**
*Meaning in life may predict Chinese undergraduates’ mental health levels through the mediating role of self-esteem.*


Interpersonal trust may also play an important role in the relationship between meaning in life and mental health. Interpersonal trust is a generalized expectation of trustworthiness that individuals place on the words and agreements of others during interpersonal interactions. It is a significant factor in maintaining and developing healthy social bonds [13]. On the one hand, interpersonal trust promotes mental health. From the perspective of individual resources, the richness of individual resources is closely related to mental health. Interpersonal trust, as a crucial component of individual resources, is a useful supplement to individual resources and can support mental health [14]. From the perspective of social networks, good interpersonal trust can facilitate the establishment and expansion of individual social networks, support them in having more constructive interactions with others, make it easier for them to access social support, and improve individual psychological well-being [15]. Extant studies have also confirmed that low levels of interpersonal trust are often associated with adverse psychological symptoms such as depression and poor health [3,16]. On the other hand, an individual’s meaning in life affects his or her interpersonal trust. Meaning in life is not only related to the individual’s own internal psychology, but also has the function of serving interpersonal relationships. It affects the way individuals treat people and things and how they behave, and it enables them to promote or improve interpersonal relationships. An individual’s motivation to find meaning in life implies a tendency to associate with those who have high meaning in life [17], and those with high meaning in life are also likely to invest more time and energy in maintaining healthy interpersonal relationships [18]. Interpersonal trust is an indicator of relationship quality and an important manifestation of interpersonal ties [19]. Therefore, individuals with a higher meaning in life may also have more positive attitudes or cognitive tendencies toward others, which, in turn, affects the individual’s level of interpersonal trust.

**Hypothesis** **2** **(H2):**
*Meaning in life may predict an individual’s mental health through the mediating role of interpersonal trust.*


There is a close relationship between an individual’s level of self-esteem and interpersonal trust; self-esteem is an individual’s attitude and perception of self, and positive self-esteem is an individual’s positive and favorable evaluation of self [20]. Self-esteem also affects individuals’ perceptions towards others and things, which, in turn, affects their interpersonal trust [21]. Previous studies have shown that not only is there a significant positive relationship between self-esteem and interpersonal trust, but also that self-esteem has a positive predictive effect on interpersonal trust [22]. Meaning in life also mobilizes inner resources such as self-esteem, promotes interpersonal relationships, and enhances the sense of happiness in life [23].

**Hypothesis** **3** **(H3):**
*There are multiple mediating roles of self-esteem and interpersonal trust in the effect of meaning in life on the mental health of Chinese undergraduates. To be specific, meaning in life may predict the mental health of Chinese undergraduates through the chain mediation of self-esteem and interpersonal trust.*


In summary, meaning in life can influence mental health, and there is a correlation among meaning in life, self-esteem, and interpersonal trust. All three have predictive effects on mental health. However, the underlying mechanisms between sense of meaning, self-esteem, interpersonal trust, and mental health remain unclear. Therefore, this study aimed to investigate the underlying process of how meaning in life, self-esteem, and interpersonal trust affect mental health.

## 2. Materials and Methods

### 2.1. Procedure

This study was approved by the Biomedical Research Ethics Review Committee of Fujian Medical University (approval No. 43, date of approval 11 March 2021). Each participant was informed about the details of this study and subsequently read and signed an informed consent form before data collection.

This is a cross-sectional study. Five different types of colleges and universities were selected in Fujian Province and randomly selected classes in each school. Cluster sampling was conducted in each class. The students in each class were invited to finish the questionnaire on the online platform Wenjuanxing (www.wjx.cn) (accessed on 30 March 2024).

This study used the Revised Meaning in Life Questionnaire to measure meaning in life, the Revised Self-Esteem Scale to measure self-esteem, the Interpersonal Trust Scale to measure interpersonal trust, and the Kessler10 to measure mental health. In addition, the scale also contains items used to collect sociodemographic data. The question “How do you feel about academic stress” was used to measure students’ academic stress. Students were asked to answer this item with mild, moderate, or severe.

### 2.2. Participants

A stratified random sampling was conducted on undergraduate students from 5 universities in Fujian Province from March to April 2022. The following is a sample size formula of the current situation study:$$n=\frac{z_{a}^{2}\cdot p\cdot (1-p)}{\delta^{2}}(a=\mathrm{significance}\;\mathrm{level},\;p=\mathrm{predicted}\;\mathrm{incidence},\;\delta =\mathrm{allowable}\;\mathrm{error})$$

The “detection rate of mental health problems” was selected as the index to calculate the sample size. According to the literature, the detection rate of mental health problems of college students is 48% [24]. At the significance level of 0.05, the allowable error of 0.05 can be achieved when the sample size is 384 according to the formula. To minimize the allowable error, a total of 1174 Chinese undergraduates were recruited from Fujian Province in the China. The inclusion criteria were as follows: (1) freshman to junior undergraduate students; (2) informed consent and voluntary participation in the research. Excluding invalidated responses, 1090 participants were retained (the rate of validation was 92.84%). The exclusion criteria were as follows: (1) recall items missing more than 20% in the questionnaire; (2) apparently regular respondents. Of this total, 466 were male, 624 were female; 331 were only children, and 759 were non-only children.

### 2.3. Measures

#### 2.3.1. Revised Meaning in Life Questionnaire (RMLQ)

The MLQ was first developed by Steger et al. [2]. In this study, the Revised Meaning in Life Questionnaire (RMLQ) by Chen Wei et al. [25] was used, for which the Cronbach’s α coefficient was 0.805. It includes two subscales, presence of meaning (PM) and search for meaning (SM), with total scores ranging from 10 to 70. The higher the score, the better the meaning in life. The Cronbach’s α coefficient of RMLQ in this study was 0.853.

#### 2.3.2. Kessler10 Scale

Kessler10 scale was first developed by Kessler and Mroczek [26]. It was used to screen the frequency of psychological symptoms in the past 4 weeks. The scale is a 10-item measure. Participants score each statement on a five-point Likert scale, with a total score ranging from 10 to 50. Higher scores indicate lower levels of mental health. The Cronbach’s α coefficient in this study was 0.919.

#### 2.3.3. Revised Self-Esteem Scale (RSES)

RSES was developed by Rosenberg. It has 10 items and was rated on a four-point scale, with higher scores indicating higher levels of self-esteem [27]. The Cronbach’s α coefficient in this study was 0.840.

#### 2.3.4. Interpersonal Trust Scale (ITS)

This scale was developed by Rotter and used to measure the reliability of others’ commitments, words, and deeds toward oneself in interpersonal relationships [28]. The scale is scored on a five-point scale. It has 25 items, with a total score ranging from 25 to 125. Higher scores indicate a higher level of trust in interpersonal interactions. In this study, the Cronbach’s α coefficient was 0.713.

### 2.4. Statistical Analyses

SPSS 23.0 was used to test common methods bias, generate descriptive statistics, and calculate correlation. Mplus 7.4 was used to test the mediation model. Harman’s single-factor test was used to perform factor analysis for all items in this study. The percentage of explained variance in the first common factor in exploratory factor analysis (EFA) was 19.08%, which was below the critical value of 40%. There were 11 factors with eigenvalues greater than 1. There were no significant covariance effects, and the data were ready for further statistical analysis.

## 3. Results

### 3.1. The Overall Mental Health of Chinese Undergraduates

The Kessler10 scale was scored, and the individual’s mental health state was classified into four levels based on the total score: 10–19 (level 1, low risk of mental illness), 20–24 (level 2, low risk of mental illness), 25–29 (level 3, high risk of mental illness), and 30–50 (level 4, high risk of mental illness) [26]. The results showed that 65.5% of Chinese undergraduates had good or better mental health, and 34.5% had poor or worse mental health (see Table 1).

### 3.2. Correlation Analysis of Investigated Variables

After taking mental health as a dependent variable and controlling gender, whether the participant was an only child, and academic stress, the results in Table 2 showed that the scores of the Meaning In Life Questionnaire, Self-Esteem Scale and Interpersonal Trust Scale had significant positive correlations, respectively; the total scores of Meaning In Life Questionnaire, Self-Esteem Scale and Interpersonal Trust Scale had significant negative correlations with the scores of the Kessler10 scale, which indicated that meaning in life, self-esteem and interpersonal trust have significant positive correlations with mental health.

### 3.3. Testing the Multiple Mediating Effects

To further clarify the relationships among Chinese undergraduates’ mental health, meaning in life, self-esteem, and interpersonal trust, the AMOS structural equation model was used to test the hypotheses. After controlling gender, whether the participant was an only child, and academic stress, this study used mental health as the dependent variable, meaning in life as the independent variable, and self-esteem and interpersonal trust as the mediating variables to construct a mediation model of self-esteem and interpersonal trust between meaning in life and mental health (see Figure 1). The fit indices of the model were as follows: χ2/df = 2.976, CFI = 0.991, TLI = 0.956, SRMR = 0.015, and RMSEA = 0.043, indicating that the mediation model had good fit indices and self-esteem and interpersonal trust acted as a chain mediator between meaning in life and mental health.

To determine the significance of the mediating effects, the Bootstrap method was used to test for multiple mediating effects, and the results revealed that the upper and lower limits of the 95% confidence intervals for each mediating path did not contain 0. This indicates that the mediating model of self-esteem and interpersonal trust between meaning in life and mental health holds. The effect size of the path mediated by self-esteem was 59.08% [B = −0.348, (−0.447, −0.250)], the effect size of the path mediated by interpersonal trust was 4.75% [B = −0.028, (−0.052, −0.005)], and the effect size of the chain mediation was 1.53% [B = −0.009, (−0.018, −0.001)]. Details of each mediated pathway and its significance are shown in Table 3.

## 4. Discussion

This study found that 34.5% of Chinese undergraduates have poor or worse mental health. At present, many Chinese undergraduates are in a sub-healthy psychological state, which may be related to the special stage of their own psychological development as well as the pressure of the general social environment. Chinese undergraduates are under various pressures such as further education, employment, social adaptation, and courtship, especially due to the impact of coronavirus disease 2019 on their original way of life and study, which leads to many new challenges and induces confusion in life, and the group of Chinese undergraduates is susceptible to various external factors, which makes them prone to psychological imbalance and instability, thus affecting their mental health level [29].

First, the results of this study showed that meaning in life can predict the mental health of Chinese undergraduates, which is consistent with the existing studies [30,31]. Both meaning in life and mental health are considered important components of a happy life, with the former having an important impact on the latter [32]. Individuals who lack meaning in life are more likely to lose direction and goals in real life and experience negative emotions such as depression and anxiety [33]. People with meaning in life experience fewer conflicts when making health-related decisions and are more inclined to self-regulate, thus maintaining a higher level of mental health [34].

Second, the study’s findings showed a significant mediating path of meaning in life–self-esteem–mental health, indicating that self-esteem plays an independent mediating role between Chinese undergraduates’ meaning in life and their level of mental health, which is consistent with the results of previous studies [35,36]; so, Hypothesis 1 was verified. When individuals are unable to find meaning in life, they tend to exhibit lower levels of self-worth. Self-esteem is generally low at this time, and different degrees of psychological problems are also more likely to arise [37]. High levels of self-esteem can help individuals maintain a more positive and stable mindset, be more confident and courageous, be more able to challenge difficulties, reduce anxiety levels when encountering difficulties, mitigate negative cognitions and poor self-evaluations, and maintain psychological balance [38]. Therefore, individuals with a higher meaning in life are able to promote the development of self-esteem and improve their mental health. Further findings revealed that the mediating path of meaning in life–interpersonal trust–mental health was also significant, confirming Hypothesis 2. Individuals are able to encounter greater meanings in life if their basic psychological needs are consistently met. If psychological needs are deprived, individuals will become unmotivated; thus, they need to establish trusting relationships with others to overcome their insecurities and fears [39]. People with a greater meaning in life have a higher level of satisfaction regarding psychological needs and possess higher interpersonal adaptability. Some studies have shown that college freshmen with higher interpersonal trust can adapt to college life faster, while their interpersonal trust can negatively predict social anxiety [40]. Therefore, interpersonal trust is a vital predictor of mental health; so, the higher interpersonal trust is, the better mental health is. From an individual perspective, interpersonal trust is an individual psychological resource, and good interpersonal trust can promote mental health; from a social perspective, good interpersonal trust facilitates interpersonal interaction and social support, both of which are beneficial to mental health. When Chinese undergraduates have a higher meaning in life, they become more adaptable, generating a sense of security, which is conducive to enhancing interpersonal trust among Chinese undergraduates and building a harmonious interpersonal pattern, both of which ultimately contribute to better mental health.

In addition, this study also discovered that there was chain mediation among meaning in life, self-esteem, interpersonal trust, and mental health. The mediation effect value accounted for 1.68% of the total effect, reflecting the significant mediation effect, supporting Hypothesis 3. This indicates that meaning in life can not only have an impact on mental health through the mediating effect of self-esteem and interpersonal trust but also increase interpersonal trust and access more psychological resources by raising an individual’s self-esteem, thus improving mental health. University students with higher levels of self-esteem have greater self-identity and a tendency to trust others, while those with lower levels of self-esteem have higher self-defense and are less likely to develop trust. Background Sociometer theory posits that self-esteem is a subjective monitor of the quality of one’s interpersonal relationships [41]. When individuals experience a low meaning in life and react negatively to interpersonal relationships, self-esteem acts as a protective factor that motivates individuals to engage in certain behaviors that uphold and restore a sense of harmony in interpersonal relationships. Thus, a high level of self-esteem helps Chinese undergraduates obtain more positive emotions in interpersonal interactions, which enhances interpersonal trust, lowers the occurrence of problem behaviors, and promotes mental health.

The above results emphasize the important role of meaning in life for the mental health of undergraduate students and establish a possible model of how meaning in life affects mental health through both internal and external pathways. Meaning in life can influence their mental health both through self-esteem, a personal factor, and through interpersonal trust, their relationship with the outside world. By recognizing the relationship between meaning in life and mental health, appropriate interventions and strategies can be developed to enhance undergraduate students’ self-esteem and interpersonal trust, and subsequently improve mental health. For undergraduate students, psychoeducation or class group counseling are effective ways to improve their sense of meaning in life [42,43]. Based on this study, universities can conduct appropriate activities to improve undergraduate students’ mental health.

## 5. Limitations and Further Research

This study has certain limitations. First, this study mainly considered factors affecting undergraduate students’ mental health related to meaning in life, including self-esteem and interpersonal trust. Although this study controlled for possible factors affecting undergraduate students’ mental health such as gender, whether the participant was an only child, and academic stress, more factors such as adverse childhood experiences, interpersonal relationships, and financial stress were not considered [1]. These factors can be taken into account in future studies.

Second, the participants of this study were all from universities in Fujian Province, which may have bias due to regional cultural differences. It is unclear whether the findings can be generalized to students from other social and cultural context. A larger study could be conducted in the future to verify the cross-cultural consistency of the findings.

At last, the data of this study were collected at the same time, and a cross-sectional study could not prove causality. In addition, the self-report measures also had biases in reporting style and subjective attitudes. Therefore, more diverse methods such as longitudinal cohort designs or randomized controlled experiments could be considered in the future to further verify the model.

## 6. Conclusions

This study extends the intrinsic mechanism between undergraduate students’ meaning in life and mental health. Undergraduate students’ mental health is significantly related to their meaning in life, and self-esteem and interpersonal trust mediate the relationship between meaning in life and mental health, suggesting that self-esteem and interpersonal trust are the key factors in the influence of meaning in life on mental health; in other words, undergraduate students who have higher levels of meaning in life are able to better satisfy their self-esteem and interpersonal trust, and have a higher level of mental health. Therefore, schools and families should strengthen humanistic care, create a favorable atmosphere, encourage and help them to pursue meaning in life, and pay attention to the cultivation of various positive psychological attributes affecting the mental health of undergraduate students so as to improve the level of mental health.

## Figures and Tables

**Figure 1 behavsci-14-00720-f001:**
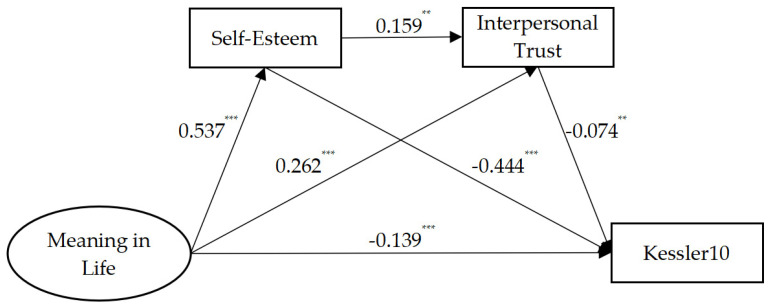
The model of self-esteem and interpersonal trust as mediators between meaning in life and mental health. ** *p* < 0.01, *** *p* < 0.001.

**Table 1 behavsci-14-00720-t001:** Overall status of Chinese undergraduates’ mental health.

Categories	N (%)	Mean	SD
10–19 (very good mental health)	422 (38.72%)	14.96	2.99
20–24 (good mental health)	292 (26.79%)	21.86	1.39
25–29 (poor mental health)	199 (18.26%)	26.95	1.41
30–50 (very poor mental health)	177 (16.24%)	33.88	4.49
Overall	1090 (100%)	22.07	7.36

**Table 2 behavsci-14-00720-t002:** Correlations of scores of Kessler10, Meaning In Life Questionnaire, Self-Esteem Scale And Interpersonal Trust Scale.

	1	2	3	4
1. Kessler10	1			
2. RMLQ	−0.318 ***	1		
3. RSES	−0.560 ***	0.408 ***	1	
4. ITS	−0.266 ***	0.248 ***	0.300 ***	1
Mean	22.07	45.68	27.94	72.92
SD	7.36	9.77	4.50	8.48

Note. Revised Meaning in Life Questionnaire (RMLQ), Revised Self-Esteem Scale (RSES), Interpersonal Trust Scale (ITS); *** *p* < 0.001.

**Table 3 behavsci-14-00720-t003:** The mediation test of self-esteem and interpersonal trust between meaning in life and mental health.

Path	Standardized Indirect Effect Value	Effect Amount	95% Confidence Interval	
			Lower Bound	Upper Bound
Total effect	−0.589	100%	−0.789	−0.389
Direct effect	−0.204	34.63%	−0.353	−0.054
Total mediated effect	−0.386	65.53%	−0.494	−0.278
RMLQ → RSES → Kessler10	−0.348	59.08%	−0.447	−0.250
RMLQ → ITS → Kessler10	−0.028	4.75%	−0.052	−0.005
RMLQ → RESE → ITS→ Kessler10	−0.009	1.53%	−0.018	−0.001

Note. Revised Meaning in Life Questionnaire (RMLQ), Revised Self-Esteem Scale (RSES), Interpersonal Trust Scale (ITS).

## Data Availability

The raw data supporting the conclusions of this article will be made available by the authors on request.

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
