# Peer review of "The Relationship between Meaning in Life and Mental Health in Chinese Undergraduates: The Mediating Roles of Self-Esteem and Interpersonal Trust"

_behavsci, 2024, doi:10.3390/bs14080720_

Round 1

Reviewer 1 Report

Comments and Suggestions for Authors

Both the study you have carried out and the article you have proposed are interesting and very timely at present.

The results obtained and conclusions  are applicable and useful in the context of interventions to promote the mental health of freshmen in schools.

I suggest you:

- Revision of the percentages shown in table 1. Overall status of Chinese Undergraduates'Mental Health - 3rd column.

- They should introduce a specific section to highlight the conclusions drawn from the results and their analysis. The most appropriate place will be after discussion and analysis of results.

Some suggestions may enrich the article and the scientific community.

Good Work.

Author Response

Dear reviewer,

We feel great thanks for your professional review work on our article. As you are concerned, there are several problems that need to be addressed. According to your nice suggestions, we have made extensive corrections to our previous draft, the detailed corrections are listed below.

Comments 1: Revision of the percentages shown in table 1. Overall status of Chinese Undergraduates' Mental Health - 3rd column.

Response: Thank you for your comment. We have already revised the format to fit the journal. If it needs further revision, please feel free to point it out.

Comments 2: They should introduce a specific section to highlight the conclusions drawn from the results and their analysis. The most appropriate place will be after discussion and analysis of results.

Response: Thank you for your suggestion. We have added this section according to the your suggestions.(see page 8 line 317 to 327)

Comments 3: Some suggestions may enrich the article and the scientific community.

Response: We appreciate your professional suggestion. We agree with your comment. We have added some strengths and appropriate suggestions according to the findings in this study.(see page 7 line 270 to 280)

Reviewer 2 Report

Comments and Suggestions for Authors

Here are the more specific comments:

The authors should include a reference to the limitations of the investigation. This includes potential biases in data collection, sample limitations, or the reliability and validity of the instruments used to measure the constructs. Without considering these limitations, it is difficult to assess the generalizability of the results. It is important to discuss the extent to which the results can be applied to other populations of students or cultural contexts.

Related to the previous point, although the discussion mentions that self-esteem and interpersonal trust act as mediators, the analysis does not sufficiently detail how these factors interact in different contexts or among different subgroups of students. The authors should deepen the analysis by including a more thorough exploration of these specific interactions. Factors such as cultural and socioeconomic contexts, or even differences between university courses or academic levels, were not considered in the analysis. These factors can significantly influence how self-esteem and interpersonal trust affect students' mental health.

The authors should include a conclusion in the article, where they synthesize the results. The lack of a final conclusion implies that there is no clear synthesis of the main findings of the study and their implications.

Round 2

Reviewer 2 Report

Comments and Suggestions for Authors

The authors made the necessary changes.